# Exploration of the ocular surface infection by SARS-CoV-2 and implications for corneal donation: An ex vivo study

Corantin Maurin[1], Zhiguo He[1], Marielle Mentek[1], Paul Verhoeven[2,3], Sylvie Pillet[2,3], Thomas Bourlet[2,3], Françoise Rogues[4], Jean Loup Pugniet[4], Thierry Peyragrosse[4], Marion Barallon[4], Chantal Perrache[1], Inès Aouimeur[1], Sophie Acquart[5], Sandrine Ninotta[5], Marc Baud'huin[6], Bertrand Vabres[6,7], Sylvain Poinard[1,8], Philippe Gain[1,8], Gilles Thuret[1,8]*

1 Laboratory "Biology, engineering and imaging of Corneal Graft" BiiGC, Faculty of Medicine, University Jean Monnet, Saint-Etienne, France, 2 CIRI, Centre International de Recherche en Infectiologie, GIMAP Team University of Lyon, University of St-Etienne, INSERM U1111, CNRS UMR5308, ENS de Lyon, UCBL1, St-Etienne, France, 3 Laboratory of Infectious Agents and Hygiene, University Hospital of St-Etienne, St-Etienne, France, 4 Hospital coordination of organ and/or tissue retrieval, University Hospital, Saint-Etienne, France, 5 Eye bank, French Blood Center, Saint-Etienne, France, 6 Nantes Université, CHU Nantes, Banque Muti-Tissus, Nantes, France, 7 Nantes Université, CHU Nantes, Service Ophthalmologic, Nantes, France, 8 Ophthalmology Department, University Hospital, Saint-Etienne, France

* gilles.thuret@univ-st-etienne.fr

**Data Availability Statement:** All relevant data are within the manuscript and its Supporting Information files.

## Abstract

### Background

The risk of Severe Acute Respiratory Syndrome Coronavirus 2 (SARS-CoV-2) transmission through corneal graft is an ongoing debate and leads to strict restrictions in corneas procurement, leading to a major decrease in eye banking activity. The aims of this study are to specifically assess the capacity of human cornea to be infected by SARS-CoV-2 and promote its replication ex vivo, and to evaluate the real-life risk of corneal contamination by detecting SARS-CoV-2 RNA in corneas retrieved in donors diagnosed with Coronavirus Disease 2019 (COVID-19) and nonaffected donors.

### Methods and findings

To assess the capacity of human cornea to be infected by SARS-CoV-2, the expression pattern of SARS-CoV-2 receptor angiotensin-converting enzyme 2 (ACE-2) and activators TMPRSS2 and Cathepsins B and L in ocular surface tissues from nonaffected donors was explored by immunohistochemistry ($n = 10$ corneas, 78 ± 11 years, 40% female) and qPCR ($n = 5$ corneas, 80 ± 12 years, 40% female). Additionally, 5 freshly excised corneas (80 ± 12 years, 40% female) were infected ex vivo with highly concentrated SARS-CoV-2 solution ($10^6$ median tissue culture infectious dose ($TCID_{50}$)/mL). Viral RNA was extracted from tissues and culture media and quantified by reverse transcription quantitative PCR (RT-qPCR) (viral RNA copies) 30 minutes (H0) and 24 hours (H24) after infection. To assess the risk of corneal contamination by SARS-CoV-2, viral RNA was tested by RT-qPCR (Ct value) in both corneas and organ culture media from 14 donors diagnosed with COVID-19 (74 ± 10

**Funding:** This study was funded by Agence Nationale de la Recherche (grant ANR-20-COV1-00, https://anr.fr/en/) to GT. The funders had no role in study design, data collection and analysis, decision to publish, or preparation of the manuscript.

**Competing interests:** The authors have declared that no competing interests exist.

**Abbreviations:** ACE-2, angiotensin-converting enzyme 2; COVID-19, Coronavirus Disease 2019; Ct, cycle threshold; IQR, interquartile range; OCT, optimal cutting temperature; PBS, phosphate buffer saline; PFA, paraformaldehyde; PVP-I, povidone-iodine; RT-qPCR, reverse transcription quantitative PCR; SARS-CoV-2, Severe Acute Respiratory Syndrome Coronavirus 2; SEM, scanning electron microscopy; TEM, transmission electron microscopy.

years, 29% female) and 26 healthy donors (79 ± 13 years, 57% female), and in organ culture media only from 133 consecutive nonaffected donors from 2 eye banks (73 ± 13 years, 29% female).

The expression of receptor and activators was variable among samples at both protein and mRNA level. Based on immunohistochemistry findings, ACE-2 was localized mainly in the most superficial epithelial cells of peripheral cornea, limbus, and conjunctiva, whereas TMPRSS2 was mostly expressed in all layers of bulbar conjunctiva.

A significant increase in total and positive strands of IP4 RNA sequence (RdRp viral gene) was observed from 30 minutes to 24 hours postinfection in central cornea ($1.1 \times 10^8$ [95% CI: $6.4 \times 10^7$ to $2.4 \times 10^8$] to $3.0 \times 10^9$ [$1.4 \times 10^9$ to $5.3 \times 10^9$], $p = 0.0039$ and $2.2 \times 10^7$ [$1.4 \times 10^7$ to $3.6 \times 10^7$] to $5.1 \times 10^7$ [$2.9 \times 10^7$ to $7.5 \times 10^7$], $p = 0.0117$, respectively) and in corneoscleral rim ($4.5 \times 10^9$ [$2.7 \times 10^9$ to $9.6 \times 10^9$] to $3.9 \times 10^{10}$ [$2.6 \times 10^{10}$ to $4.4 \times 10^{10}$], $p = 0.0039$ and $3.1 \times 10^8$ [$1.2 \times 10^8$ to $5.3 \times 10^8$] to $7.8 \times 10^8$ [$3.9 \times 10^8$ to $9.9 \times 10^8$], $p = 0.0391$, respectively). Viral RNA copies in ex vivo corneas were highly variable from one donor to another. Finally, viral RNA was detected in 3 out of 28 corneas (11%) from donors diagnosed with COVID-19. All samples from the 159 nonaffected donors were negative for SARS-CoV-2 RNA. The main limitation of this study relates to the limited sample size, due to limited access to donors diagnosed with COVID-19 and concomitant decrease in the procurement corneas from nonaffected donors.

## Conclusions

In this study, we observed the expression of SARS-CoV-2 receptors and activators at the human ocular surface and a variable increase in viral RNA copies 24 hours after experimental infection of freshly excised human corneas. We also found viral RNA only in a very limited percentage of donors with positive nasopharyngeal PCR. The low rate of positivity in donors diagnosed with COVID-19 calls into question the utility of donor selection algorithms.

## Trial registration

Agence de la Biomédecine, PFS-20-011 https://www.agence-biomedecine.fr/.

## Author summary

### Why was this study done?

- Corneal transplantation is by far the most common transplantation procedure in the world, and there is a severe lack of corneal donation to respond to the increasing patient need.

- The persistent Coronavirus Disease 2019 (COVID-19) pandemic has dramatically worsened this situation by limiting the access to corneas during a long period of the pandemic and finally increasing the long list of contraindications to corneal donation based solely on an exacerbated precautionary principle.

- Exploration of the Severe Acute Respiratory Syndrome Coronavirus 2 (SARS-CoV-2) risk of transmission through corneal transplantation is mandatory to adapt the donor

selection strategy and improve corneal tissue procurement in the persisting COVID-19 crisis.

## What did the researchers do and find?

- We explored and observed the expression of 4 proteins related to molecular pathways allowing SARS-CoV-2 cellular entry at the ocular surface (cornea, bulbar conjunctiva, 10 samples). These proteins were differentially located in the studied tissues, and their expression was variable among donors.

- The experimental infection of 5 freshly excised corneas with highly concentrated SARS-CoV-2 viral solution promoted only moderate and variable viral RNA multiplication in corneal samples. Twenty-four hours after inoculation, a significant increase in total and positive strand viral RNA was observed in the epithelium of central cornea (p-value: 0.0039 for and 0.0117, respectively) and in corneoscleral rim (p-value: 0.0039 and 0.0391, respectively).

- We explored the presence of SARS-CoV-2 viral genetic material in corneas and respective storage media from donors tested positive or negative to COVID-19 PCR on nasopharyngeal swab. All samples from the 159 donors not contaminated by COVID-19 were negative for SARS-CoV-2 RNA. SARS-CoV-2 RNA was detected in only 3 corneas out of 28 corneas (11%) from donors diagnosed with COVID-19.

## What do these findings mean?

- Our results support that the ocular surface expresses in a variable way the proteins involved in SARS-CoV-2 cellular entry and infection. We observed a moderate and variable increase in total and positive viral RNA after experimental infection of human freshly excised corneas. A very low rate of positivity in donors diagnosed with COVID-19 was observed, supporting a low risk of SARS-CoV-2 presence in donor corneas.

- This work suggests that it would be beneficial to procure corneas from donors tested positive to COVID-19 (nasopharyngeal swabs), test their corneal storage media, and transplant the negative corneas.

## Introduction

The very nature of the ocular surface, composed of tissues of different embryological origins, and its obvious exposure to aerosols poses 2 unresolved questions in this period of persistent Coronavirus Disease 2019 (COVID-19) pandemic: Is there a risk of transmission of the Severe Acute Respiratory Syndrome Coronavirus 2 (SARS-CoV-2) via corneal transplantation? Is the ocular surface a route of entry and exit for the virus?

Formally eliminating the hypothesis of SARS-CoV-2 transmission by corneal transplant would allow the suppression of additional PCR testing and associated extra costs that extend the already long list of microbiological tests performed, and some of which not based on a clearly identified transmission risk, as for toxoplasmosis and syphilis. It should be noted that documented cases of transmission of systemic infectious diseases to the recipient through

keratoplasty can be counted on the fingers of one hand for hundreds of thousands of transplants carried out in more than a century [1,2]. Increasing the long list of contraindications to the use of donor corneas will exacerbate the already existing global shortage of donor tissue, where one donor is available for 70 patients in need [3]. Additionally, it impedes the work of hospital coordination for tissues and organs procurement, brings confusion, and increases the risk of staff discouragement. Finally, it deprives some patients and/or families of access to tissue donation, which can be a beneficial aspect of the mourning process, already deeply modified by the pandemic.

In March 2020, exclusion of donors who tested positive for the SARS-CoV-2 was also based on the need to protect retrieval and technical staff from eye banks, as inadequate or absent robust data on SARS-CoV-2 transmission were available. Following this principle of precaution, deceased persons were systematically isolated in watertight body bags. Nevertheless, these recommendations became obsolete at least in France in November 2020 [4,5] because (1) there is no production of aerosols or infectious droplets from deceased persons; (2) no infectious droplets are projected during cornea procurement; (3) the general principle of precaution applied for all organ procurement (surgical mask + eye protective equipment) is sufficient to protect the staff; and (4) finally, there is no risk of transmission during corneal handling by the eye bank staff. Thus, the strict restrictions applied following the principle of precaution, liable to a major impact on grafting delay for patients, should now be replaced by evidence-based recommendations. To mention, eye banking activity in Europe underwent a major drop in activity during the first period of pandemic (March to May 2020), most of them related to drastic donor selection measures than to the severity of the pandemic [6].

The possibility of transmission via the conjunctival route, mentioned at the beginning of the pandemic [7], has only been explored in vivo in 2 rhesus macaques. After ocular conjunctival inoculation (topically), they developed a moderate form of COVID-19 without any ocular symptoms and viral RNA was detected in conjunctival swabs 1 day after infection but not thereafter. A nasolacrimal drainage of the virus leading to the respiratory tract infection represents the most probable contamination pathway in this experiment [8]. Several studies reported that viral RNA was inconsistently detected in ocular surface samples (conjunctival swabs or tear samples) of confirmed COVID-19 cases [9]. Positivity rate in ocular surface samples of COVID-19 patients is likely dependent on the stage of the disease at the time of corneal retrieval and the retrieval technique [10]. Likewise, highly variable prevalence of ocular lesions in patients affected by SARS-CoV-2 are also reported, ranging from 0.8% to 31.6% [11–12]. Thus, the available data provide only limited and conflicting evidence to support a possible SARS-CoV-2 infection via the corneal or conjunctival route.

Entry of SARS-CoV-2 in host cells requires the specific interaction between its transmembrane spike (S) glycoprotein and the host-cell surface receptor angiotensin-converting enzyme 2 (ACE-2) [13], followed by the cleavage of its Spike (S) glycoprotein by host proteases, such as transmembrane serine protease TMPRSS2 [14] or endosomal cysteine proteases Cathepsin B and Cathepsin L [15], leading to virus–cell fusion and intracellular viral RNA release. While evidence of ACE-2 and TMPRSS2 expression in corneal epithelial cells was previously provided [16–18], their expression in bulbar conjunctival epithelium was not observed at the protein nor the mRNA level and must be further confirmed [19]. The expression in corneal and conjunctival epithelium of Cathepsins B and L, implicated in alternative viral entry pathways, remain hypothetical. Cathepsin B was described only in the epithelial cells and keratocytes of keratoconic corneas [20].

For corneal transplantation, donor corneas are always procured with a scleral rim and therefore contain residues of conjunctiva even if this one is largely resected. The presence of contaminating viruses in corneas stored in eye banks is still under debate. Bayyoud and

colleagues found no viral RNA in fresh (unstored) corneas from 5 patients with confirmed COVID-19 who died from respiratory insufficiency [21]. In this study, decontamination with povidone-iodine (PVP-I) or no decontamination of the corneas were performed prior to tissue retrieval in 2 independent groups and resulted in similar results. On the contrary, 2 other studies reported positive viral detection in corneas of patients with confirmed COVID-19. Sawant and colleagues, studying 10 donors with confirmed COVID-19 that underwent ocular PVP-I disinfection prior to tissue procurement, found viral RNA in 3 conjunctiva from 2 donors, 1 swab of the corneal surface and 5 swabs of the endothelial surface [22]. Casagrande and colleagues reported the presence of genomic viral RNA in 6 out of 11 corneal tissue (11 patients) without adjacent sclera or conjunctiva and procured without disinfection of the ocular surface [23].

In the present study, we provide new clues to answer these questions with 3 complementary experimentations. First, we analyzed the expression of SARS-CoV-2 receptors and activators at the ocular surface. Secondly, we explored whether SARS-CoV-2 could infect freshly excised corneas ex vivo. Finally, we searched for viral RNA in corneoscleral tissues and organ culture media from a large series of donors diagnosed with COVID-19 or not affected by COVID-19, at different times of storage.

## Material and methods

### Ethical considerations

The study complied with the Declaration of Helsinki guidelines for research involving human tissues and French Eye Bank regulations. An authorization for the collection of donors diagnosed with COVID-19 for scientific purposes was obtained from the Agence de la Biomédecine (PFS-20-011), and an authorization to leave the body bag open to allow the collection to be carried out before its final closure was obtained from the medical examiner of the Agence Régionale de Santé Auvergne Rhône Alpes. All donors' families provided informed consent for research prior to procurement. Donor tissues were retrieved at the University Hospital of Nantes and Saint-Etienne. This study is reported as per the Strengthening the Reporting of Observational Studies in Epidemiology (STROBE) guideline (S1 Checklist).

In this study, patients who died without presenting clinical signs of COVID-19 and had negative PCR results on postmortem nasopharyngeal swabs were considered as not affected and are referred to "non-COVID-19 donors" in the technical sections of the article. Patients with a positive PCR on postmortem nasopharyngeal swab were selected as contaminated patients and are referred in the text to "COVID-19 donors".

In an effort to lower any risk of bias, all samples related to COVID-19 status (donor samples from eye banks and for research purpose, corneal samples infected ex vivo) were analyzed in a blinded fashion.

### Study 1: Expression of ACE-2, TMPRSS2, Cathepsin B, and Cathepsin L in the different ocular surface tissues

**Donors and sampling procedure.** Expression of ACE-2, TMPRSS2, Cathepsin B, and Cathepsin L was studied by immunohistochemistry in corneas of non-COVID-19 donors. Ten donor corneas from 6 men and 4 women with a mean age of 78 ± 10 years were used. The corneas of 3 donors were sampled and analyzed for the expression of ACE-2 and TMPRSS2 prior to the study start. The donors included during COVID-19 pandemic ($n$ = 7) presented a negative PCR on postmortem nasopharyngeal swabs and no clinical signs related to COVID-19 before death. Corneas were procured at a mean time of 11 ± 6 hours after death following

standard procedure without disinfection to allow preservation of epithelial layers, and immediately fixed in paraformaldehyde (PFA) 0.5% during 45 minutes, followed by inclusion in optimal cutting temperature (OCT) compound (Tissue-Tek, Sakura Finetek USA, Torrance, CA, USA) and storage at −20˚C until use.

Expression of the markers of interest was also studied in conjunctival epithelial cells collected by impression cytology. A total of 11 conjunctival epithelial samples were collected in healthy donors (Mean age 53 ± 24 years, 55% female, patients undergoing cataract surgery and presenting healthy cornea and conjunctiva) by impression cytology using a dedicated device (Eyeprim OS1, OPIA Technologies, Paris, France) and immediately fixed at room temperature during 15 minutes in 0.5% PFA. Each of the patient had a negative PCR 24 to 48 hours prior to surgery and were asymptomatic. They were informed about the procedure and provided written informed consent for research purposes.

**Immunohistochemistry.**   To validate high-quality and reproducible immunolabeling of the targets under study [24], several commercial antibodies targeting the same proteins were screened on noninfected VERO E6 cells, known to express SARS-CoV-2 receptors and activators [25], and infected VERO E6 cells for the screening of antibodies directed toward viral spike and nucleocapsid proteins. Antibodies providing the strongest labeling intensity associated with accurate subcellular localization were finally used in this study to explore ACE-2, TMPRSS2, Cathepsin B, and Cathepsin L expression (S1 Table, S1 Fig). This selection was performed to avoid false negative results. Similarly, to limit erroneous positive observations, negative control conditions using mouse and/or rabbit IgG in replacement of the tested antibodies were also evaluated on the same cells and ocular tissues targeted in this study. The antibodies were further explored in conjunctival epithelial cells collected by impression cytology and flat mounted corneas.

Complete procedure and materials are detailed in Section A of S1 File and in S1 Table).

Immunostaining was performed in 10-µm thick frozen sections without prior permeabilization following the protocol detailed in the Section A of S1 File. Primary and secondary antibodies were all diluted at 1/500 and 1/1,000, respectively. Fluorescence was imaged using inverted epifluorescence microscope IX81 (Olympus, Tokyo, Japan) and confocal fluorescence microscope confocal FLUOVIEW FV1200 (Olympus, Tokyo, Japan), both equipped with FV10-ASW4.1 image analysis software. In order to facilitate precise localization of the 3 epithelial regions of cornea, limbus, and conjunctiva, a double immunostaining was performed by combining the studied markers with Cytokeratin 12, a marker of mature corneal epithelial cells (sc-515882, Santa Cruz Biotechnologies, Dallas, USA).

**Quantitative polymerase chain reaction.**   Expression of ACE-2, TMPRSS2, Cathepsin B, and Cathepsin L genes was studied by qPCR in the epithelial cells of corneal samples from ex vivo infection procedure (samples harvested 30 minutes after viral infection [H0]). The corneas used in this procedure were procured from 5 non-COVID-19 donors as described below (study 2). Cells harvested after ex vivo infection were lysed in 2 mL Nuclisens lysis buffer (Biomérieux, Marcy l'Etoile, France) during 15 minutes prior to automated RNA extraction in nucleic acid extraction system Nuclisens EMAG (Biomérieux) using dedicated reagents, cDNA was synthetized from total RNA by reverse transcription with ProtoScript II First Strand cDNA Synthesis Kit (E6560, New England BioLabs). Reverse transcription was conducted during 1 hour at 42˚C followed by enzyme inactivation at 80˚C during 5 minutes. Quantitative PCR was performed using Luna Universal qPCR Master Mix Kit (M3004, New England BioLabs) and PCR primers (Sigma-Aldrich, St. Louis, MO, USA) for the following human genes: Glyceraldehyde-3-Phosphate Dehydrogenase (GAPDH), ACE-2, TMPRSS2, Cathepsin B, and Cathepsin L. The list of primers was provided in S2 Table. The following qPCR protocol was performed: initial denaturation at 95˚C during 60 seconds followed by 50 cycles of denaturation at 95˚C (15 seconds) and annealing/extension at 58˚C (30 seconds).

Every sample was tested in duplicate during 2 distinct runs, and milliQ water was used as negative control. Results were expressed as $2^{-\Delta Ct}$ with adjustment for amplification efficiency [26].

## Study 2: Ex vivo experimental infection of fresh corneas with SARS-CoV-2

**Donors and sampling procedure.** To further study the possibility of SARS-CoV-2 infection of the ocular surface, we evaluated viral multiplication capacity in freshly excised human corneal and scleral tissues (limbus and sclera with bulbar conjunctiva). The replication of SARS-CoV-2 was quantified by qPCR, researching the IP4 fragment of the *RdRp* gene specific to SARS-CoV-2.

Corneas were procured in 5 non-COVID-19 scientific donors (mean age 80 ± 12 years, 40% female) within 6 hours after death without decontamination with PVP-I in order to preserve an intact ocular surface as much as possible. The 5 donors presented a negative PCR on postmortem nasopharyngeal swab and no clinical signs related to COVID-19 before death. After checking that corneas were transparent without epithelial ulceration, a thorough rinse with Balanced Salt solution was performed and the corneas were retrieved with a 4-mm scleral rim taking care of letting the bulbar conjunctiva intact. Corneas were incubated for 30 minutes in Corneamax (Eurobio, Les Ulis, France) supplemented with 1/100 antibiotic antimycotic solution (A5955, Sigma Aldrich) prior to processing.

**Infection procedure.** All procedures were performed in the Biological Safety Level 3 platform (BSL3) of EA3064 –GIMAP laboratory at Faculty of Medicine of Saint-Etienne. The viral strain (RoBo strain) used in this study was a clinical isolate obtained from a nasopharyngeal aspirate of a patient hospitalized at the University Hospital of Saint-Etienne for severe COVID-19 and cultured on Vero-E6 cells (CRL-1586, ATCC) [27]. The viral particles were stored as a stock solution at −150°C in liquid nitrogen at a viral load of $10^8$ median tissue culture infectious dose (TCID50)/mL. After thawing, stock solution was serially diluted and used to infect Vero-E6 cells ($10^6$ cells/ml) in 96 wells plates during 5 days at 37°C and 5% $CO_2$. After the incubation phase, cytopathogenic effect on culture cells as checked and viral load was evaluated as previously described prior to infection [28].

In order to eliminate potential replication in the endothelial cells, identified target cells for SARS-CoV-2 entry and replication [16,17,29], they were removed using a surgical microsponge and efficacy of scrapping controlled by trypan blue staining (the Descemet membrane then turns completely blue). Corneas were then placed on a 3D printed curved support with epithelium upward to allow cutting in several parts without damaging the surface (Fig 1B). We analyzed separately the central part of the cornea corresponding to the transplanted part, and the periphery, which is almost never used (quite exceptionally for limbal allografts and arciform grafts for peripheral ulcerations). First, the center of the cornea was trephined using an 8.25-mm diameter punch. Secondly, the central cornea and the peripheral corneoscleral rim (comprising corneal periphery, limbus, and sclera covered by conjunctiva) were further cut into 3 and 5 pie-shaped pieces, respectively. Each tissue part (8 per cornea: 5 corneoscleral rim and 3 central cornea portions) was then covered with a nonwoven compress moistened with 150 μL of $10^6$ $TCID_{50}$/mL SARS-CoV-2 viral solution. Incubation was performed in a humid chamber at 37°C and 5% $CO_2$ for 1 hour. After incubation, compresses were carefully removed from each tissue piece, which were then rinsed 3 times with phosphate buffer saline (PBS) to remove free viruses. Each sample was then placed in 6-well culture plates using 700 μL (central cornea) or 1,000 μL (corneoscleral rim) of Corneamax (Eurobio) in order to maintain the epithelial surface mainly in contact with air and prevent cell shedding in the culture medium (Fig 1B). For each pair of corneas, tissue pieces from one cornea were incubated at 37°C in a 5% $CO_2$ incubator for 30 minutes (hereafter referred as H0) and tissue pieces from the other cornea for 24 hours (H24). Thirty-minute incubation theoretically ensured maximal viral

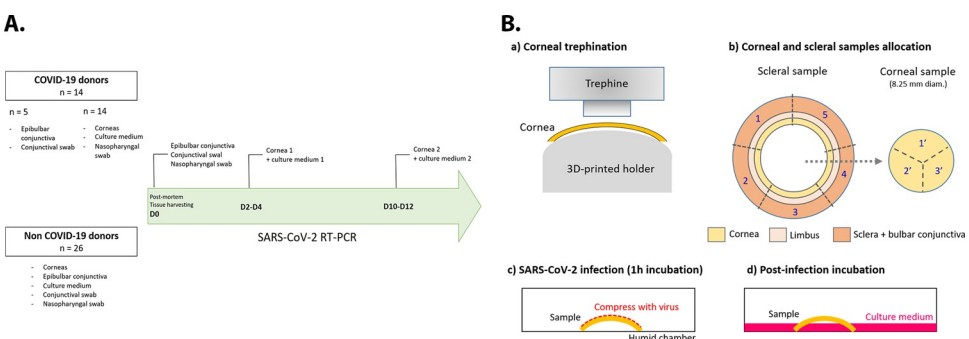

**Fig 1.** Schematic representation of the experimental procedures performed for the detection of SARS-CoV-2 RNA in the different ocular samples of COVID-19 and non-COVID-19 donors (A), and for the ex vivo experimental infection of fresh corneas with SARS-CoV-2 (B). COVID-19, Coronavirus Disease 2019; RT-PCR, reverse transcription PCR; SARS-CoV-2, Severe Acute Respiratory Syndrome Coronavirus 2.

adhesion and cell infection but limited replication, whereas 24-h incubation allowed the observation of viral replication [30]. After incubation, the epithelial cells of each tissue piece and the corresponding culture medium were collected for either viral RNA detection or qPCR. Whole tissue pieces were also collected for immunohistolabeling.

**RT-PCR detection for SARS-CoV-2 RNA.** Cells harvested after ex vivo infection were lysed in 2 mL Nuclisens lysis buffer (Biomérieux), during 15 minutes prior to automated RNA extraction in nucleic acid extraction system Nuclisens EMAG (Biomérieux) using dedicated reagents. GAPDH housekeeping gene expression was used to normalize the number of cells analyzed for viral RNA detection.

Viral RNA was detected by targeting IP4 sequence in RdRp gene following Pasteur Institute recommendations. Total viral RNA was quantified by one-step reverse transcription quantitative PCR (RT-qPCR) using SuperScript III Platinum (Invitrogen, Thermo Fisher) following Pasteur Institute protocol [31] with minor modifications: reverse transcription at 55°C during 15 minutes, polymerase activation and initial denaturation at 95°C during 2 minutes followed by 50 cycles of denaturation at 95°C (15 seconds) and annealing/extension at 58°C (30 seconds). The sequences of primers were as follows: Forward primer: 5′-GGTAACTGGTAT GATTTCG-3′; Reverse primer: 5′-CTGGTCAAGGTTAATATAGG-3′; Fluorescent probe: 5′- TCATACAAACCACGCCAGG-Fam-BHQ-1-3′.

Total viral RNA (IP4), positive (IP4+), and negative (IP4−) RNA strands were amplified in separated runs: Both primers were used for the detection of total viral RNA, whereas only forward or reverse primers were added to the reaction mix for the detection of positive and negative RNA strands. In that case, reverse transcription was performed in a first separate step with either forward (negative strand) or reverse primer (positive strand) using ProtoScript II First Strand cDNA Synthesis Kit (E6560, New England BioLabs).

Negative RNA strands represent specifically the amount of replication matrix present in the cells, whereas positive strands represent both the matrix used for transcription (subgenomic mRNA) and the virion genome. As the RdRp gene is located near the 5′ extremity of SARS-CoV-2 genomic RNA, the probability of subgenomic mRNA amplification is very low. Based on this hypothesis, we approximated the increase in positive strand RNA with an increase in viral genomic RNA. Reverse transcription was conducted during 1 hour at 42°C followed by enzyme inactivation at 80°C during 5 minutes. Quantitative PCR was finally performed on the obtained total, positive strands and negative strands cDNA using Luna Universal qPCR Master Mix Kit (M3004, New England BioLabs) following the steps: initial denaturation at 95°C during 60 seconds followed by 50 cycles of denaturation at 95°C (15 seconds) and annealing/

extension at 58˚C (30 seconds). Standard Quantification method was applied to quantify viral RNA expression (expressed as cycle threshold (Ct) values). The maximal number of PCR cycles observed for the viral RNA amplification was 40. Ct values ranged from 20 to 30 for total IP4, from 20 to 38 for IP4+ strand, and from 25 to 40 for IP4− strand.

The whole-genome SARS-CoV-2 RNA was used as a positive control (Sequence from eurogentec: 5′-GGTAACTGGTATGATTTCGGTGATTTCATACAAACCACGCCAGGTAGT GGAGTTCCTGTTGTAGATTCTTATTATTCATTGTTAATGCCTATATTAACCTTG ACCAG-3′) while negative control consisted in replacement of template by sterile water (10977035, Invitrogen).

To support the results obtained with RdRp gene, we also performed another qPCR targeting both the N gene and RdRp gene on the remaining samples (ex vivo infection media: $n = 9$ samples for H0 and $n = 9$ samples for H24), using a commercial kit (ARGENE SARS-CoV-2 R-GENE kit, #423735, Biomérieux) [32].

**Viral particles detection.**　To detect SARS-CoV-2 particles at the cell surface, both transmission (TEM) and scanning electron microscopy (SEM) were performed on infected (H0 – viral adhesion) or noninfected tissues. Detailed experimental procedure is provided in Section B of S1 File. In addition, immunostaining of SARS-CoV-2 particles was performed after 24 hours of infection following the procedure detailed in Section B of S1 File.

**Assessment of replicable SARS-CoV-2 virus presence.**　To explore the production of infectious viral particles, we cultivated Vero-E6 cells in the presence of organ culture media (Cornea-Max) sampled from the ex vivo infections. We incubated 80% confluent Vero-E6 cells with 200 μl of organ culture media sampled at H0 ($n = 9$ H0 central cornea media and $n = 9$ H0 corneoscleral rim media) and H24 ($n = 9$ H24 central cornea media and $n = 9$ H24 corneoscleral rim media). A total of 36 samples were thus tested. Negative control cells were incubated with sterile culture medium, whereas positive control cells were incubated with the viral solution used for the infection procedure (undiluted stock solution: $10^8$ TCID50/mL and dilution at 1/100: $10^6$ TCID50/mL). Cells were incubated 7 days at 37˚C and 5% $CO_2$ and daily visually checked to observe the presence of cytopathogenic effect caused by SARS-CoV-2 multiplication. At the end of incubation period, cells were then fixated in 0.5% PFA and immunolabeled against SARS-CoV-2 nucleocapsid protein (40150 R007, SinoBiological), nuclei were stained with TO-PRO-3 Iodide (T3605, Invitrogen). Cells were observed by confocal laser scanning microscopy (FV1200, Olympus).

## Study 3: SARS-CoV-2 RNA detection in the ocular surface tissues from COVID-19 and non-COVID-19 donors

**Donors and corneal retrieval procedure.**　SARS-CoV-2 RNA detection was performed on corneas from COVID-19 and non-COVID-19 donors included between May 2020 and April 2021 (10 months). Postmortem nasopharyngeal swab was systematically performed on all donors at the time of death. Corneas were procured by in situ corneoscleral excision following standard eye bank procedure that include PVP-I disinfection. Disposable surgical instruments were used and were disinfected between the procurement of the right and left eyes. Corneas were immersed in 100 mL of culture medium (Corneamax) and stored in sealed flasks at 31˚C as for standard eye banking.

The 14 COVID-19 (mean age $74 \pm 10$ years, 29% female) donors were patients with a positive PCR on postmortem nasopharyngeal swab. Five donors were for donation for scientific purposes and presented compatible clinical signs combined with a positive PCR result on nasopharyngeal swab at $9 \pm 16$ days before death. The remaining 9 cases were asymptomatic donors procured for therapeutic use after selection following the strict algorithm in force in France during the study period [6]. They nevertheless presented a positive PCR on

postmortem nasopharyngeal swabs. The 2 corneas (and respective culture medium) from one donor were analyzed separately and randomly on the 2 different days of the analysis: One cornea and its culture medium were analyzed between 2 and 4 days (D) after procurement (D2 group), and the second cornea (with culture medium) was analyzed 8 days later (D10 group), corresponding to the mean storage time in European eye banks.

The 159 non-COVID-19 donors were patients who died without signs of COVID-19 (asymptomatic) and had negative PCR results on postmortem nasopharyngeal swabs. In 133 eye bank donors (therapeutic use, mean age 73 ± 13 years, 29% female), the organ culture medium was sent for analysis at the end of storage. In the remaining 26 non-COVID-19 donors procured for scientific purposes (mean age 79 ± 13 years, 58% female), both corneas and culture media were analyzed.

In total, samples were consecutively procured from 142 (133 non-COVID-19 + 9 COVID-19) consecutive donors for therapeutic use following the selection algorithm in force in France during the study period, and 31 (26 non-COVID-19 + 5 COVID-19) donors for scientific purposes. In addition to corneas, the epibulbar conjunctiva and a conjunctival swab were also collected in the 31 donors for scientific purpose. The detailed study flow chart processing was summarized in Fig 1A.

**RT-PCR detection for SARS-CoV-2 RNA.** Total RNA from the entire corneas and culture media (200 μl) of COVID-19 and non-COVID-19 donors was extracted following manufacturer's instruction using either RNeasy mini-Kit (Qiagen, Hilden, Germany) or Quick-RNA kit (Zymo Research, Irvine, USA) depending on kit availability fluctuations at the time of experiments. Consistency between the 2 extraction kits was experimentally validated prior to sample processing (Section C of S1 File and S6 Table). Total viral RNA was detected as described above.

## Statistical analysis

Sample size calculation was performed using G-power V3.1 as previously described [33] and using available SARS-Cov-2 in vitro infection data [34]. We approximated the standard deviation at half the mean of the number of viral RNA copies at H0 and H24 ($2.5.10^7$ and $2.5 \times 10^8$ copies/mL, respectively). With $\alpha = 0.05$ and $\beta = 0.95$, we obtained an effect size of 1.8869 and a total sample size of 5.

Analysis was performed using GraphPad Prism (GraphPad Software, La Jolla, CA, USA). The normality of the distribution of continuous variables was tested by the Shapiro–Wilk normality test. Continuous variables with normal distribution were reported as mean ± standard deviation (SD); nonnormal variables were reported as median (interquartile range (IQR)). Experimental groups were either compared by paired *t* test (normal distribution) or Wilcoxon signed-rank test (nonnormal distribution). Correlation between gene expression of the various receptor/activators and total IP4 level at H0 was studied using Spearman correlation coefficient. $p < 0.05$ was considered significant.

## Results

### Expression of ACE-2, TMPRSS, Cathepsin B, and Cathepsin L in the different ocular surface tissues

ACE-2 was expressed in the most superficial cells from the different epithelia of the ocular surface in 10/10 donors. A moderate interindividual variability was noted in the limbus and conjunctiva, with a labeling observed in all corneas (10/10 samples), in most of the limbus (9/10), and in the conjunctiva of 8 out of 10 donors. In the 8 donors expressing ACE-2 on the whole

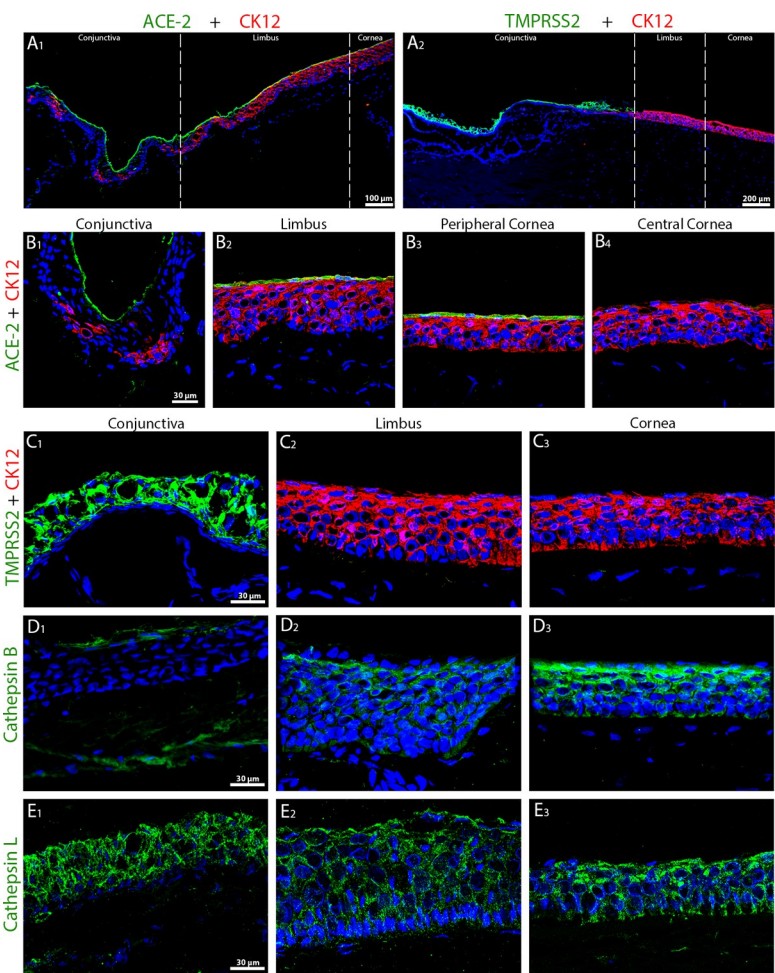

**Fig 2. Representative immunostaining of ACE-2, TMPRSS2, Cathepsin B, and Cathepsin L in cryosections of 0.5% PFA-fixed fresh human corneas.** A1 and A2, low-magnification immunostaining of ACE-2 or TMPRSS2, respectively (green, Alexa 488), and CK12 (red, Alexa 555) in sections of the limbal region showing the transition from peripheral cornea to bulbar conjunctiva. B1–B4, immunohistostaining of ACE-2 (green, Alexa 488) and CK12 (red, Alexa 555) in conjunctiva, limbus, peripheral and central cornea. C1–C3, immunohistostaining of TMPRSS2 (green, Alexa 488) and CK12 (red, Alexa 555) in conjunctiva, limbus, and cornea. D1–3 and E1–3, immunohistostaining of Cathepsin B (D) or Cathepsin L (E) (both green, Alexa 488) in conjunctiva, limbus, and cornea. Nuclei were stained using Hoechst-33342 (blue). ACE-2, angiotensin-converting enzyme 2; CK12, cytokeratin 12; PFA, paraformaldehyde.

ocular surface, we observed a progressive decrease in the immunofluorescence signal from the conjunctiva to the corneal center, where ACE-2 signal was almost absent (Fig 2A1 and 2B1-4).

TMPRSS2 was highly expressed in all cell layers of the conjunctival epithelium (Fig 2A2 and 2C1-3) compared to limbus and corneal epithelium where signal was almost absent. A similar expression pattern was found in 7 fresh corneas out of 9. In 2 donors, TMPRSS2 was not expressed.

Cathepsin B was expressed only by 2 donors out of 6: in 1 case, in the conjunctiva (epithelium and substantia propria) (Fig 2D1) and in another case in all layers of limbal and corneal epithelium (Fig 2D2-3). Cathepsin L was expressed by 5 out of 7 donors, with significant variability between tissues: 5 times in the cornea, but only 1 time in the limbus and 2 times in the conjunctiva. In each case, all epithelial cell layers were labeled (Fig 2E1-32). The 2 tissues negative for Cathepsin L were also negative for Cathepsin B.

Immunostaining of superficial conjunctival epithelial cells, harvested by impression cytology in healthy donors, are detailed in Section B in S1 File and confirmed immunohistochemistry results obtained on fresh corneas (S2 Fig). Moreover, the subcellular localization of each marker could be distinguished, with TMPRSS2 (S2A Fig) and ACE-2 localizing at the cellular membrane and Cathepsins B and L (S2B Fig) presenting a cytoplasmic localization. In addition, a few CD45+ immune cells were present in corneal epithelium and the conjunctival epithelium and expressed ACE-2 and TMPRSS2, respectively (S3 Fig).

SARS-CoV-2 receptors and activators gene expression was studied by qPCR in 5 samples of central cornea and corneoscleral rims. We also found great variability in the amount of ACE-2, TMPRSS2, Cathepsin L, and Cathepsin B mRNA in the central cornea and corneoscleral rim of the 5 donors (Fig 3): In corneoscleral rims, TMPRSS2 showed the highest variability followed by ACE-2, Cathepsin L and Cathepsin B genes. In the central cornea, Cathepsin B showed the highest variability followed by ACE-2, Cathepsin L, and TMPRSS2. We found no correlation between gene expression of the 4 receptor/activators and total IP4 level at H0 (Spearman correlation coefficient, $p > 0.05$).

Taken together, these results strongly suggest the significant presence of both SARS-CoV-2 host-cell surface receptor ACE-2 and major actors of S protein priming, TMPRSS2, Cathepsin B, and Cathepsin L, in epithelial cells of the ocular surface tissues, with variable localization and degree of expression. In addition, these results suggest a variable expression of these proteins among individuals.

## Ex vivo experimental infection of fresh corneas with SARS-CoV-2

After 24 hours of infection, total IP4 copy number increased significantly from $1.1 \times 10^8$ (IQR: $8.4 \times 10^7$ to $1.7 \times 10^8$, 95% CI: $6.4 \times 10^7$ to $2.4 \times 10^8$) to $3.0 \times 10^9$ (IQR: $9.1 \times 10^8$ to $6.0 \times 10^9$, 95% CI: $1.4 \times 10^9$ to $5.3 \times 10^9$, $p = 0.0039$) in central corneas, and from $4.5 \times 10^9$ (IQR:

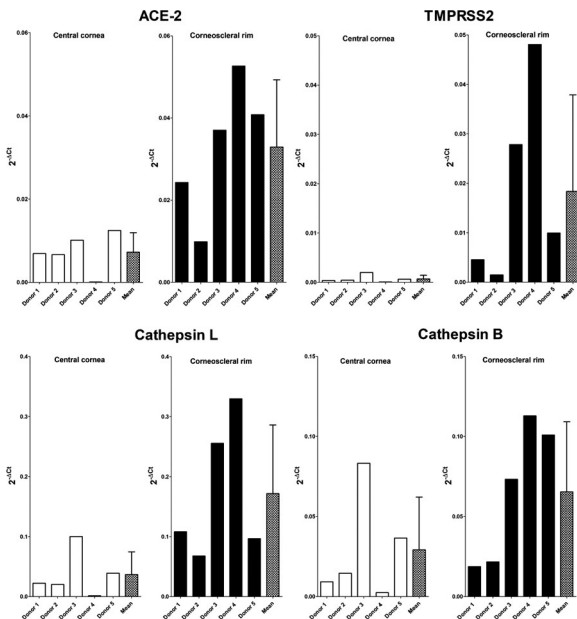

**Fig 3. Expression of ACE-2, TMPRSS2, Cathepsin B, and Cathepsin L genes in the epithelium of ex vivo infected central cornea and corneoscleral rim at H0 ($n$ = 5 samples).** Results are expressed as $2^{-\Delta Ct}$ as no control group was used in this experiment. Mean ± standard deviation of the 5 samples is provided in the right-end bar. ACE-2, angiotensin-converting enzyme 2.

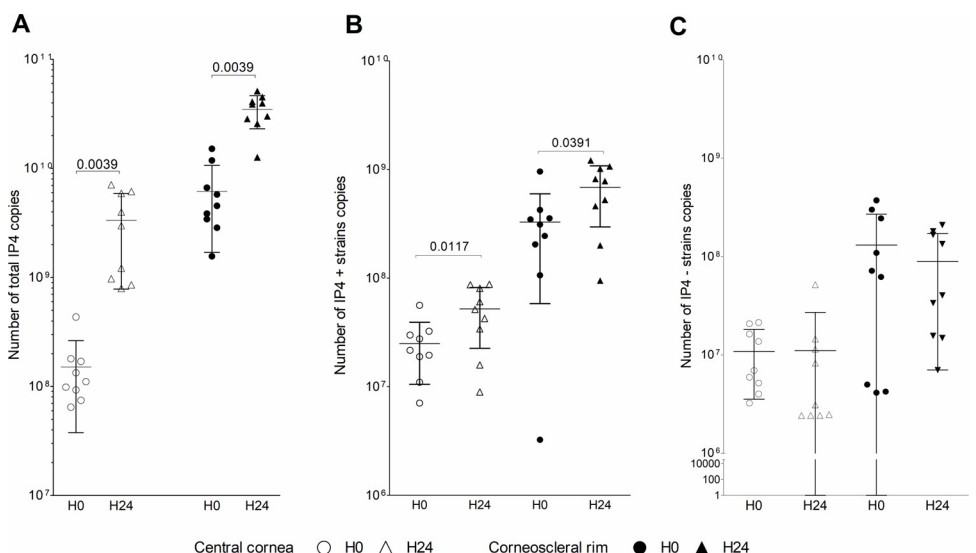

**Fig 4.** SARS-CoV-2 RNA expression as median ± IQR number of total IP4 copies (A), of IP4 positive copies (IP4+, B) and IP4 negative copies (IP4−, C) in the epithelium of central cornea and corneoscleral rim at 30 minutes (H0) and 24 hours (H24) after SARS-CoV-2 ex vivo infection. Results are provided for the epithelium of central cornea and corneoscleral rim pooled with their respective culture medium. Wilcoxon signed-rank test for paired samples was used to compare H0 and H24 data. IQR, interquartile range; SARS-CoV-2, Severe Acute Respiratory Syndrome Coronavirus 2.

$3.1 \times 10^9$ to $9.2 \times 10^9$, 95% CI: $2.7 \times 10^9$ to $9.6 \times 10^9$) to $3.9 \times 10^{10}$ (IQR: $2.7 \times 10^{10}$ to $4.3 \times 10^{10}$, 95% CI: $2.6 \times 10^{10}$ to $4.4 \times 10^{10}$, $p = 0.0039$) in corneoscleral rims.

The copy number of the positive IP4 strand increased also significantly in central corneas from $2.2 \times 10^7$ (IQR: $1.5 \times 10^7$ to $3.1 \times 10^7$, 95% CI: $1.4 \times 10^7$ to $3.6 \times 10^7$) to $5.1 \times 10^7$ (IQR: $2.5 \times 10^7$ to $8.4 \times 10^7$) (95% CI: $2.9 \times 10^7$ to $7.5 \times 10^7$, $p = 0.0117$) and in corneoscleral rims from $3.1 \times 10^8$ (IQR: $1.5 \times 10^8$ to $3.9 \times 10^8$, 95% CI: $1.2 \times 10^8$ to $5.3 \times 10^8$) to $7.8 \times 10^8$ (IQR: $3.3 \times 10^8$ to $1.0 \times 10^9$, 95% CI: $3.9 \times 10^8$ to $9.9 \times 10^8$, $p = 0.0391$).

Variations did not reach significance level for the IP4 negative strands copies, which were highly variable among donors (Fig 4). Fold changes in IP4 total number of copies between H0 and H24 for each donor were represented in S4 Fig. The replication of SARS-CoV-2 ex vivo was highly variable from one donor to another and particularly important in the central cornea with fold changes (%) ranging from 3% to 92%, while it ranged from 2% to 20% in corneoscleral rims. Comparable results were obtained with the amplification of both RdRp and N genes in culture media using a commercially available kit (S5 Fig).

Vero-E6 cells cultured in the presence of both pure and diluted viral solution used for the infection procedure (positive controls) presented cytopathogenic effects after 3 days of incubation, whereas negative control cells remained intact. Additionally, all cells presenting cytopathogenic effects were positive for N-protein expression. One well of cells cultured with tested samples was positive for N-protein expression, whereas no cytopathogenic effects was observed and was thus considered as positive for viral multiplication.

The first cytopathogenic effects on cells incubated with ex vivo samples were observed after 4 days of incubation. Five over 9 H24 central cornea media and 3 over 9 H24 corneoscleral rim media promoted viral multiplication in Vero-E6 cells. Among these 8 H24 positive samples, 4 of them were associated with a paired H0 medium promoting cytopathogenic effects, suggesting the presence of infecting viruses from the initial infection solution at H0 and a possible persistence of these viruses after 24 hours of incubation.

The results of SEM and TEM imaging of infected and noninfected corneal and corneoscleral samples are detailed in S6 Fig and did not allow us to differentiate viral particles from microplicae and microvilli at the surface of epithelial cells. Similarly, immunostaining of viral particles was not conclusive as nonspecific labeling was detectable in noninfected tissues and only low labeling of viral particles was observed in infected tissues (S7 Fig).

## SARS-CoV-2 RNA detection in the ocular surface tissues from COVID-19 and non-COVID-19 donors

COVID-19 donors' characteristics and their RT-PCR results were shown in Table 1. Notably, among the 14 COVID-19 donors, 9 were donors for therapeutic use, thus were selected as presenting no history of clinical signs of COVID-19, with a positive PCR on postmortem nasopharyngeal swab despite selection algorithm. They represent 6.3% (9 over 142) of the total consecutive asymptomatic donors included during the study period. Viral RNA was detected after 2 days of storage only in 1 cornea and its culture medium, and in 2 other corneas (1 cornea alone and 1 cornea with its organ culture medium) after 10 days of storage. These 5 positive samples were from asymptomatic donors retrieved for therapeutic use. No bilateral contamination was found. RT-PCR were all negative in postmortem conjunctival samples (epibulbar tissue and swabs).

The 159 non-COVID-19 and asymptomatic donors were 98 men and 61 women with a mean age of 73.7 ± 14 years (min 19, max 103). Corneas were procured 15 ± 8.6 hours after death. The mean duration of organ culture was 19 ± 5.4 days. RT-PCR of all samples from this series were negative (52 corneas, 52 epibulbar conjunctival swabs and tissue, 318 (266 + 52) organ culture media).

**Table 1. Characteristics of the 14 COVID-19 donors and SARS-CoV-2 PCR results in their corneas and the associated organ culture media after 2 and 10 days of storage in standard eye banking conditions.** Day 2 corresponds to one cornea and day 10 to the other one of the same patient. Ct values were provided for positive results.

| Donation purpose | Age (years) | Gender | Death to procurement time (h) | Symptoms of COVID-19 | Day 2 | | Day 10 | |
|---|---|---|---|---|---|---|---|---|
| | | | | | PCR cornea | PCR culture medium | PCR cornea | PCR culture medium |
| Scientific | 87 | F | 8.7 | Yes | (−) | (−) | (−) | (−) |
| | 65 | M | 27.5 | Yes | (−) | (−) | (−) | (−) |
| | 82 | M | 0.5 | Yes | (−) | (−) | (−) | (−) |
| | 57 | F | 17 | Yes | (−) | (−) | (−) | (−) |
| | 67 | F | 23 | Yes | (−) | (−) | (−) | (−) |
| Therapeutic | 68 | M | 9.7 | No | (−) | (−) | (−) | (−) |
| | 91 | M | 15.7 | No | **38 Ct** | **35 Ct** | (−) | (−) |
| | 85 | F | 9.5 | No | (−) | (−) | (−) | (−) |
| | NA | M | 10.25 | No | (−) | (−) | **37.5 Ct** | (−) |
| | 72 | M | 8.5 | No | (−) | (−) | (−) | (−) |
| | 72 | M | 6.75 | No | (−) | (−) | **34 Ct** | **33 Ct** |
| | 63 | M | NA | No | (−) | (−) | (−) | (−) |
| | 79 | M | NA | No | (−) | (−) | (−) | (−) |
| | 69 | M | 5.5 | No | (−) | (−) | (−) | (−) |
| Mean ± SD | 74 ± 10 | | 11 ± 7 | | | | | |

COVID-19, Coronavirus Disease 2019; Ct, cycle threshold; F, female; M, male; NA, not available; SARS-CoV-2, Severe Acute Respiratory Syndrome Coronavirus 2; SD, standard deviation.

## Discussion

In the present study, SARS-CoV-2 RNA replication was detected in fresh human cornea and conjunctiva 24 hours after viral infection, both tissues expressing SARS-CoV-2 receptor and activators ACE-2, TMPRSS2, and Cathepsin B/L with variable amount and localization. Meanwhile, SARS-CoV-2 RNA detection in corneas was negative in the general population and remained very rare in cases diagnosed with COVID-19.

We were able to characterize the expression of ACE-2 and the 3 viral entry cofactors, TMPRSS2, Cathepsin B, and Cathepsin L in different ocular surface locations at both proteins and transcripts levels. Our findings complete the previously reported data in human corneas [16–18,35,36] with a detailed and robust cartography of each protein expression pattern. Superficial epithelial cells from corneal periphery, limbus, and from bulbar conjunctiva were the main sources of ACE-2 expression, the key receptor binding SARS-CoV-2 at the cell surface. While ACE-2 expression was mainly limited to superficial cells, TMPRSS2 was expressed in the whole thickness of the epithelium in bulbar conjunctiva and almost absent from corneal epithelium. Thus, the central part of the cornea, which is the one that is transplanted in more than 99% of cases, may not be a good viral vector. As far as we know, we describe for the first time the expression pattern of Cathepsins B and L proteins and mRNA at the healthy human ocular surface while the expression of both proteins was only explored in human whole corneas by qPCR and western blot [37] and Cathepsin B in human keratoconus corneal samples [20].

The variable expression of the targeted proteins among our samples may suggest a variable degree of expression of these proteins at the ocular surface among the general population. This observation was particularly pronounced for Cathepsins B and L, showing a highly variable expression limited to only a few donors. Thus, these results would support a very variable individual sensitivity to viral contamination through the ocular surface route.

To the best of our knowledge, we documented for the first time SARS-CoV-2 RNA replication in freshly excised and intact human corneas 24 hours after inoculation with the virus. We chose an inoculum of $10^6$ TCID$_{50}$/mL SARS0-CoV-2, corresponding to $3.7E^{10} \pm 3.1E^7$ copies/ml in the experimental solution at T0 and a contact time of 30 minutes. We chose a long incubation time, much longer than the contact time of a Flugge droplet on the ocular surface, as an extreme situation (worst case, mimicking for instance a repeated exposition). Moreover, the absence of lacrimal secretions containing proteins with antimicrobial activity like lactoferrin should further favor viral activity. On the contrary, we chose an inoculum consistent with the maximal viral RNA load reported in throat swab and sputum samples of patients 5 to 6 days after symptoms onset [38]. This viral load is also in accordance with the only animal experiment available, reporting the development of mild interstitial pneumonia in 2 out of 2 rhesus monkeys inoculated once with SARS-CoV-2 via conjunctival route [8]. Interestingly, Miner and colleagues, using a similar viral load of $10^6$ PFU/mL with unspecified incubation time, found no viral replication in 7 human corneas (discarded from an eye bank), previously stored at 4˚C in Optisol-GS for an unspecified duration [39]. It is very likely that standard disinfection procedures before procurement followed by variable storage duration in a short-term storage medium at a nonphysiological temperature strongly decreased the survival of epithelial cells [40] or, at least, altered their protein expression profile. Taken together, both studies point out that only intact ocular surface epithelia are permissive for SARS-CoV-2. In addition, Makovoz and colleagues showed active replication of SARS-CoV-2 in human primary cultures of limbal and corneal epithelia (more in the limbus than in the cornea) [41].

Despite very favorable conditions for the viral contamination of the ocular surface, with prolonged and direct contact with viral particles, SARS-CoV-2 RNA replication after 24 hours

was moderate, with total and positive viral RNA showing a significant increase. These results support a capacity of SARS-CoV-2 RNA to replicate in ocular surface cells ex vivo; however, the production of new infectious viral particles was not proved. Indeed, the results of the infection of Vero-E6 cells with the organ culture media from the ex vivo infection showed little evidence of production of infectious viral particles during the ex vivo infection, with half of the H24 media without any signs of viral multiplication.

Moreover, the viral RNA amount quantified by qPCR was variable among samples. This observation emphasized the highly probable role of the variable expression of viral entry receptor ACE-2 and cofactors TMPRSS2, Cathepsin B, and Cathepsin L at the ocular surface between corneal samples, contributing to differential individual susceptibility to viral contamination through ocular route. In addition, the negative RNA strand unprotected by viral particle is more sensitive to rapid degradation during experiment, contributing to higher variability and higher Ct values observed in quantitative results.

Despite the absence of SARS-CoV-2 RNA in the corneas from a first report [21], viral RNA has been detected in a few corneas procured in donors diagnosed with COVID-19 following standard retrieval procedures. Our findings of SARS-CoV-2 RNA detected in 3 corneas out of 28 (19 donors) confirm previous results of a low rate of contamination of corneas procured in donors diagnosed with COVID-19 [22]. As previously reported by Sawant and colleagues, bilateral virus detection was likewise not systematic in our cohort, with only one positive cornea per donor. Interestingly, Casagrande and colleagues recently reported the presence of viral RNA in a much higher number of corneal discs (without scleral rim) procured without disinfection procedure from autopsies, with genomic RNA detected in 6 samples out of 11 [23].

To the best of our knowledge, we analyzed for the first time the presence of SARS-CoV-2 RNA during standard organ culture, the prevalent storage method in Europe. Viral RNA presence was checked at 2 time points, with last analysis after 10 to 12 days of storage, which represent a classical storage duration. For the 14 donors tested positive to COVID-19, only 2 corneas were positive after 10 to 12 days of storage, supporting a persistence of detectable viral genomic material under these conditions. The detection of viral genomic material in culture medium in 2 samples out of 28 from these donors suggest the release of viral genomic material and/or the desquamation of infected cells in the culture medium and support the pertinence of analyzing these samples. However, RT-qPCR technique is not able to discriminate between intact viral RNA able to replicate and altered RNA material. Thus, it is not possible to conclude on the replication capacity of the SARS-CoV-2 RNA detected in these samples. The well-described progressive deterioration of corneal epithelium during eye bank storage [42] does not seem to represent favorable conditions to allow massive viral replication. Similarly, by combining progressive deterioration of corneal epithelium [40] and decline in metabolic activity, cornea storage at low temperature (4˚C) may represent even less favorable conditions for viral replication. Finally, despite reports of SARS-CoV-2 genetic material in corneas of donors diagnosed with COVID-19, no studies have demonstrated that this viral genomic material from corneal samples is able to promote secondary infection of another tissue.

In our cohort, all samples from the 165 donors tested negative to COVID-19 were negative for SARS-CoV-2 RNA. Up to now, the absence of viral detection in ocular tissues from patients not affected by COVID-19 was only reported in 4 donors [22].

The main limitation of this study relates to the limited sample size, due to limited access to donors diagnosed with COVID-19 and concomitant decrease in the procurement corneas from nonaffected donors [6]. Notably, despite repeated information to the care teams, their extra workload, the fear of leaving the body bag open, and the organization of the rapid departure of bodies to private mortuaries did not allow all possible donors to be procured. In particular, the limited sample size did not allow exploring possible cofactors associated with the risk

of contamination. Another limitation relates to the incapacity of our experiments to provide clear clues in favor of the presence of infectious viral particles 24 hours after infection of the cornea. Thus, the observed increase in viral genetic material can be linked to an abortive process or the production of infectious viral particles in ocular surface cells.

The highly favorable conditions for viral contamination used in our experiment cannot be extrapolated to a real-life situation mainly because of the extended contact time we used. Consequently, the possibility, in the general population, of contamination of the respiratory tract exclusively through the ocular surface, the lacrimal tract, and then the nasopharynx remains hypothetical. In practice, for such a contamination to be demonstrated, it would be necessary to prove that an individual wearing an effective mask was exposed to the virus solely by the ocular route, in order to exclude a concomitant direct and indirect contamination. Corneal transplantation, on the other hand, is a specific situation where the ocular route could be the only route of SARS-CoV-2 contamination. It would then have to be demonstrated that the central cornea can contain a sufficient quantity of active viruses after the usual processing and storage by an eye bank and that the lacrymo-rhino-pharyngo-respiratory route is effective for SARS-Cov-2 infection in this particular context.

The absence of detectable viral genomic material in samples (organ culture medium and/or cornea) from preselected asymptomatic donors with negative nasopharyngeal PCR highlights the limited usefulness of systematic PCR testing of storage media on such donors. In contrast, the few traces of SARS-CoV-2 RNA in the cornea and/or organ culture medium were from asymptomatic donors with positive nasopharyngeal PCR and none from symptomatic donors. These results call into question the relevance of donor selection algorithms.

Our results are not in favor of a significant replication of SARS-CoV-2 during eye banking, supporting an improbable risk of viral transmission through keratoplasty. On the other hand, the risk linked to a persistent visual handicap due to a lack of donor corneas is evident and increased by measures that are now unfounded. It is now necessary to question the relevance of continuing to systematically exclude deceased donors diagnosed with COVID-19 and to carry out a study to objectively measure whether there is a risk of transmission. Restriction of PCR tests on the storage medium only for the donors with positive nasopharyngeal PCR should be implemented to our standard operating processes. Our results suggest that only corneas with positive PCR in their storage medium should be excluded, pending further studies to demonstrate whether they are able to transmit the virus.

In conclusion, ACE-2 receptor and TMPRSS2 activator concentrated in corneal periphery and bulbar conjunctiva, combined with more variable expression of Cathepsins B and L in the whole cornea, were associated with variable and moderate replication of SARS-CoV-2 virus after experimental infection of freshly excised human corneas. In real life, the detection of viral RNA was very limited in corneas from donors tested positive to COVID-19, at the beginning or during standard organ culture.

## Supporting information

**S1 Checklist. STROBE checklist.**
(DOCX)

**S1 File. Supplemental methods and results. (A)** Immunohistochemistry protocols validation. (**B**) Ex vivo experimental infection of fresh corneas with SARS-CoV-2. (**C**) SARS-CoV-2 RNA detection in the ocular surface tissues from COVID-19 and non-COVID-19 donors. COVID-19, Coronavirus Disease 2019; SARS-CoV-2, Severe Acute Respiratory Syndrome Coronavirus 2.
(DOCX)

**S1 Fig. Immunostaining of targeted proteins in VERO E6 cells.** Representative immunostaining results obtained with the selected panel of primary antibodies targeting ACE-2, TMPRSS2, Cathepsin B, and Cathepsin L in VERO E6 cells (A) and targeting viral proteins Spike 3525 and Capside 40143 on VERO E6 cells infected by a clinical strain of SARS-CoV-2 (B). Immunostaining was tested on both methanol and PFA-fixed cells. ACE-2, angiotensin-converting enzyme 2; PFA, paraformaldehyde; SARS-CoV-2, Severe Acute Respiratory Syndrome Coronavirus 2.
(TIF)

**S2 Fig. Representative examples of ACE-2, TMPRSS2, Cathepsin B, and Cathepsin L positive immunostaining on conjunctival and limbal superficial epithelial cells retrieved by impression cytology in healthy donors.** A colabeling with CK12 (expressed in mature corneal and limbal epithelial cells) confirmed that the present sample contained almost exclusively conjunctival epithelial cells. (**A**) Low-magnification view of a complete impression membrane showing ocular localization of the TMPRSS2 (green, Alexa 488) and CK12 (red, Alexa 555) proteins in superficial epithelial cells (cornea and conjunctiva). The localization of these markers is further shown at high magnification in images A1 to 3. (**B**) Immunostaining of ACE-2, Cathepsin B, and Cathepsin L (green, Alexa 488) at low (top panel) and high magnification with orthogonal view (lower panel). ACE-2, angiotensin-converting enzyme 2; CK12, cytokeratin 12.
(TIF)

**S3 Fig. Immunostaining of ACE-2, TMPRSS2, and CD45 receptor on flat mounted cornea.** Immunostaining of ACE-2 or TMPRSS2 (green, Alexa 488) and CD45 receptor (red, Alexa 555) in the corneal epithelium (A) and conjunctival epithelium (B) from a flat mounted cornea of a healthy donor. ACE-2, angiotensin-converting enzyme 2.
(TIF)

**S4 Fig. Fold changes in SARS-CoV-2 IP4 total number of copies between H0 and H24 for each donor in central cornea and corneoscleral rim.** Results are given for the epithelium of central cornea and corneoscleral rim pooled with their respective culture medium. SARS-CoV-2, Severe Acute Respiratory Syndrome Coronavirus 2.
(TIF)

**S5 Fig. SARS-CoV-2 RNA expression in ex vivo infection.** SARS-CoV-2 RNA expression as mean Ct ± SD of Cov-IP4-RdRp-2020 PCR (A), of Cov-N-2021 PCR (B) and of Cov-RdRp-2021 PCR (C) in the incubation media of central cornea and corneoscleral rim at 30 minutes (H0) and 24 hours (H24) after SARS-CoV-2 ex vivo infection. Results are provided for the respective culture medium of central cornea and corneoscleral rim. H0 and H24 data were compared using a paired *t* test. Ct, cycle threshold; SARS-CoV-2, Severe Acute Respiratory Syndrome Coronavirus 2.
(TIF)

**S6 Fig. SEM and TEM exploration of SARS-CoV-2 viral particles at the ocular surface.** Representative SEM (A) and TEM (B) images of the limbal epithelial surface of infected and noninfected (control) limbal tissue. SARS-CoV-2, Severe Acute Respiratory Syndrome Coronavirus 2; SEM, scanning electron microscopy; TEM, transmission electron microscopy.
(TIF)

**S7 Fig. Immunostaining of SARS-CoV-2 proteins.** Immunostaining of viral proteins Capside 40143 (green, Alexa 488) and CK12 (red, Alexa 555) in central cornea and corneoscleral rim, 24 hours after SARS-CoV-2 infection. Immunostaining was performed on flat mounted

corneas. (**A**) Viral capsid protein labeling, shown here at increasing magnification, was concentrated in CK12 nonlabeled cells (conjunctival epithelium) and was similar to labeling observed in vitro in VERO E6 cells, whereas the labeling was more sporadic in both corneal periphery and center. (**B**) Nevertheless, at higher magnification, SARS-CoV-2 capsid protein labeling had a similar pattern in both infected tissues and controls, highlighting a possible nonspecific labeling of viral protein. CK12, cytokeratin 12; SARS-CoV-2, Severe Acute Respiratory Syndrome Coronavirus 2.
(TIF)

**S1 Table. List of primary antibodies validated on VERO-6 cells and the secondary antibodies used for immunohistochemistry on all studied samples.**
(DOCX)

**S2 Table. Sequences of primers used for quantitative RT-PCR study of SARS-CoV-2 receptor and activators.** RT-PCR, reverse transcription PCR; SARS-CoV-2, Severe Acute Respiratory Syndrome Coronavirus 2.
(DOCX)

**S3 Table. Raw data of the expression of ACE-2, TMPRSS2, Cathepsin B, and Cathepsin L genes in the epithelium of ex vivo infected central cornea and corneoscleral rim at H0 ($2^{-\Delta Ct}$ values) (Fig 3).** ACE-2, angiotensin-converting enzyme 2.
(DOCX)

**S4 Table. Raw data of SARS-CoV-2 RNA expression as mean number of total IP4 copies, of IP4 positive copies (IP4+) and negative copies (IP4−) in the epithelium of central cornea and corneoscleral rim at 30 minutes (H0) and 24 hours (H24) after SARS-CoV-2 ex vivo infection (Fig 4) are provided in the tables below.** SARS-CoV-2, Severe Acute Respiratory Syndrome Coronavirus 2.
(DOCX)

**S5 Table. Corresponding raw Ct values (S5 Fig).** Ct, cycle threshold.
(DOCX)

**S6 Table. Ct values obtained by RT-qPCR for the panel of samples extracted with the two extraction kits.** Ct, cycle threshold; RT-qPCR, reverse transcription quantitative PCR.
(DOCX)

## Acknowledgments

The authors wish to thank Cyrille Haddar, Josselin Rigaill, and Estelle Audoux from GIMAP laboratory for their technical support, and Nora Mallouk-Forges from the common center of electron microscopy of Jean Monnet University for her expertise and technical support. The authors also wish to thank the Agence Nationale de la Recherche (ANR) for the funding of this project.

## Author Contributions

**Conceptualization:** Zhiguo He, Paul Verhoeven, Sylvie Pillet, Thomas Bourlet, Marc Baud'huin, Bertrand Vabres, Philippe Gain, Gilles Thuret.

**Formal analysis:** Corantin Maurin, Zhiguo He, Marielle Mentek, Chantal Perrache, Inès Aouimeur, Sylvain Poinard.

**Funding acquisition:** Paul Verhoeven, Thomas Bourlet, Marc Baud'huin, Bertrand Vabres, Philippe Gain, Gilles Thuret.

**Investigation:** Corantin Maurin, Zhiguo He, Françoise Rogues, Jean Loup Pugniet, Thierry Peyragrosse, Marion Barallon, Chantal Perrache, Inès Aouimeur, Sophie Acquart, Sandrine Ninotta.

**Methodology:** Corantin Maurin, Zhiguo He, Marielle Mentek, Paul Verhoeven, Sylvie Pillet, Thomas Bourlet, Philippe Gain, Gilles Thuret.

**Resources:** Paul Verhoeven, Sylvie Pillet, Thomas Bourlet, Françoise Rogues, Jean Loup Pugniet, Thierry Peyragrosse, Marion Barallon, Sophie Acquart, Sandrine Ninotta, Marc Baud'huin, Bertrand Vabres, Sylvain Poinard.

**Supervision:** Zhiguo He, Marielle Mentek.

**Writing – original draft:** Corantin Maurin, Zhiguo He, Marielle Mentek, Sylvain Poinard, Gilles Thuret.

**Writing – review & editing:** Corantin Maurin, Zhiguo He, Marielle Mentek, Paul Verhoeven, Sylvie Pillet, Thomas Bourlet, Françoise Rogues, Jean Loup Pugniet, Thierry Peyragrosse, Marion Barallon, Chantal Perrache, Inès Aouimeur, Sophie Acquart, Sandrine Ninotta, Marc Baud'huin, Bertrand Vabres, Sylvain Poinard, Philippe Gain, Gilles Thuret.

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
