## [Editor Report · Decision Letter 0]

3 Jun 2021

Dear Dr Thuret, 

Thank you for submitting your manuscript entitled "Experimental studies of the ocular surface contamination by SARS-CoV-2 and implications for corneal donation" for consideration by PLOS Medicine.

Your manuscript has now been evaluated by the PLOS Medicine editorial staff and I am writing to let you know that we would like to send your submission out for external peer review.

Please re-submit your manuscript within two working days, i.e. by Jun 07 2021 11:59PM.

Kind regards,

Caitlin Moyer, Ph.D.

Associate Editor

PLOS Medicine

---

## [Decision Letter · Decision Letter 1]

26 Oct 2021

Dear Dr. Thuret,

Thank you very much for submitting your manuscript "Experimental studies of the ocular surface contamination by SARS-CoV-2 and implications for corneal donation" (PMEDICINE-D-21-02421R1) for consideration at PLOS Medicine. 

Your paper was evaluated by a senior editor and discussed among all the editors here. It was also sent to three independent reviewers, including a statistical reviewer. The reviews are appended at the bottom of this email and any accompanying reviewer attachments can be seen via the link below:

[LINK]

As you will see, the reviewers have raised a number of substantial concerns regarding the methodology and interpretations of the results. I am afraid that we will not be able to accept the manuscript for publication in the journal in its current form, but we will consider a revised version that thoroughly addresses all of the reviewers' and editors' comments. Obviously we cannot make any decision about publication until we have seen the revised manuscript and your response, and we plan to seek re-review by one or more of the reviewers. 

We expect to receive your revised manuscript by Nov 16 2021 11:59PM. Please email us (plosmedicine@plos.org) if you have any questions or concerns.

We look forward to receiving your revised manuscript. 

Sincerely,

Caitlin Moyer, Ph.D.

Associate Editor

PLOS Medicine

plosmedicine.org

1. Requests of Reviewers: Please completely address all points raised by reviewers. In particular, please address Reviewer 2’s request to clarify/share the SARS-CoV-2 isolation/culture methods, the verification of ex vivo infection using an additional virus-specific sequence, the question of potential for false positive results and whether tissue contains replicable virus, and please address Reviewer 3’s point that the data do not support the conclusion that replication of SARS-CoV-2 in tissue was demonstrated.

2. Title: Please revise your title according to PLOS Medicine's style. Your title must be nondeclarative and not a question. It should begin with main concept if possible. "Effect of" should be used only if causality can be inferred, i.e., for an RCT. Please place the study design ("A randomized controlled trial," "A retrospective study," "A modelling study," etc.) in the subtitle (ie, after a colon).

3. Data Availability Statement: Please revise the data statement, which now reads: “All relevant data are within the manuscript and its Supporting Information files.” 

PLOS Medicine requires that the de-identified data underlying the specific results in a published article be made available, without restrictions on access, in a public repository or as Supporting Information at the time of article publication, provided it is legal and ethical to do so. Please see the policy at 

http://journals.plos.org/plosmedicine/s/data-availability

and FAQs at 

http://journals.plos.org/plosmedicine/s/data-availability#loc-faqs-for-data-policy

PLOS defines the “minimal data set” to consist of the data set used to reach the conclusions drawn in the manuscript with related metadata and methods, and any additional data required to replicate the reported study findings in their entirety. Authors do not need to submit their entire data set, or the raw data collected during an investigation. However, we request that you please make available the values behind the means, standard deviations and other measures reported, any values derived from images, and the values used to build graphs.

For each data source used in your study: 

4. Throughout: Please include line numbers with the revised version.

5. Throughout: We suggest referring to “donors diagnosed with COVID-19” or similar rather than “COVID19 donors” or “COVID-19 deceased donors” throughout the text.

6. Abstract: Methods and Findings: Please indicate the setting of the study and describe the cornea donor population/source of donor tissue and clearly note outcome measures for each part of the study.

7. Abstract: Methods and Findings: Please quantify the main results (with 95% CIs and p values).

8. Abstract: Conclusions: Please address the study implications without overreaching what can be concluded from the data; beginning with the phrase "In this study, we observed ..." may be useful.

9. Author summary: We ask that you include a short, non-technical Author Summary of your research to make findings accessible to a wide audience that includes both scientists and non-scientists. The Author Summary should immediately follow the Abstract in your revised manuscript. This text is subject to editorial change and should be distinct from the scientific abstract. Please see our author guidelines for more information: https://journals.plos.org/plosmedicine/s/revising-your-manuscript#loc-author-summary

10. Methods: Study 1: Please describe the relevant demographics of the 10 cornea donors who did not have COVID-19, and please describe how it was established that they did not have COVID-19. Please describe how these were sampled, and why numbers of IHC samples differed for each marker. For the conjunctival epithelial samples, please describe how the 11 healthy donors were selected, and how it was established that they did not have COVID-19.

11. Methods: Study 2: Please describe how it was established the 5 donors did not have COVID-19.

12. Discussion: Please present and organize the Discussion as follows: a short, clear summary of the article's findings; what the study adds to existing research and where and why the results may differ from previous research; strengths and limitations of the study; implications and next steps for research, clinical practice, and/or public policy; one-paragraph conclusion.

13. Discussion: “We describe for the first time the expression pattern of cathepsins B and L proteins and mRNA at the healthy human ocular surface…” Please temper all claims of primacy with “To the best of our knowledge” or similar.

14. S1 Table: Where label intensity is described, please clarify “?” for SARS-CoV-2 labelling in cornea.

15. Reporting Guidelines: Please ensure that the study is reported according to the STROBE guideline, or the most relevant guideline for your study, and include the completed STROBE checklist as Supporting Information. When completing the checklist, please use section and paragraph numbers, rather than page numbers. Please add the following statement, or similar, to the Methods: "This study is reported as per the Strengthening the Reporting of Observational Studies in Epidemiology (STROBE) guideline (S1 Checklist)."

Please report your study according to the relevant guideline, which can be found here: http://www.equator-network.org/

Comments from the reviewers:

Reviewer #1: The aim of this study is to assess the capacity of human cornea to be infected by SARSCoV-2 and promote its replication ex vivo, and to evaluate the real-life risk of corneal contamination by detecting SARS-CoV-2 RNA in corneas and associated tissues retrieved in COVID-19 and non-COVID-19 donors. 

Comments:

Did the authors undertake an a priori sample size calculation?

"Data are expressed as Mean ± Standard Deviation (SD). Analysis was performed using GraphPad Prism (GraphPad Software, La Jolla, CA, USA). IP4 total number of copies were compared among groups using paired-T-test. Correlation between gene expression of the various receptor/activators and total IP4 level at H0 was studied using Spearman correlation coefficient."

Can the authors please confirm if the assumption of normality has been verified?

Overall, this manuscript presents an important and timely hypothesis generation study (due to the limited sample size, which the authors appropriately acknowledge in the discussion).

Reviewer #2: The authors present a study addressing the question of the possibility of a transmission of SARS-CoV-2 via donor corneas. The study is methodologically extensive and deals first with the presence of specific SARS-CoV-2 receptors in corneal and conjunctival tissue, second with the question of whether corneal and conjunctival tissue can be infected with SARS-CoV-2 ex vivo, and third with the occurrence of SARS -CoV-2 positive donor corneas and medium in clinical routine.

The question, if SARS-CoV-2 transmission can take place via corneal transplantation, is of very high importance. Especially regarding the worldwide shortage of transplants and the continuation of the activities of the corneal banks in times of the pandemic. The present study adds valuable information to the current debate. 

However, the study in its present form falls short of clearly supporting the conclusions drawn. To enhance the present paper to a publishable level, additional experimental data should be included, as well as a thorough revision of the manuscript. 

Major points: 

- Why didn't the authors employ a commercially available kit of SARS-CoV-2? The present approach makes this study essentially irreproducible for other groups and thus possibly does not meet PLOS ONE's publication criteria as outlined on the journal's website. Please make this isolation & culture method public using the supporting information section. Which variant of SARS-CoV-2 was isolated and successfully cultured?

- The proof of a successful infection ex vivo was only provided via the RdRp gene. The authors should use at least one additional virus-specific sequence (e.g. nucleocapsid protein gene or S-gene) for PCR detection to support the result. SEM and TEM images as well as immunostaining did not. 

- The nomenclature of contamination and infection in this study is confusing (e.g. page 5, para 3: "Study 2"). In the abstract it is stated, that "the aim of the study is … capacity of human cornea to be infected by SARS-CoV-2 …". As I understand it, it is infection of corneal tissue and not contamination in Study 2. Please specify and change the manuscript accordingly. 

- P 7, para 2: A Ct value of 40 as threshold value is unusually high leading to a potential high false positive rate. As the authors rightly point out in the discussion (page 16, para 1), the question of replicable viruses in corneal tissue is crucial for a potential risk to corneal recipients. The authors did not test, if the infected tissue (Study 2) contained replicable virus, this should be additionally performed. 

- Table 1: How do the authors explain the fact, that two of the three PCR positive cornea and culture medium were tested negative in Day2 and positive in Day10. In this case I would be very reluctant to speak of clear positive virus detection, especially considering the Ct value.

- P 6 para 1: The authors scraped endothelial cells before infection with SARS-CoV-2 and maintained air contact of epithelial surface during culture. The authors explained the reasons for this setting; however, this does not meet the usual corneal culture conditions in clinical routine. The authors should at least add a control group with intact endothelium layer and epithelium covered with medium. 

Minor Points: 

- P 4, para 2: Change "Bayoud et al" (Ref 19) to "Bayyoud et al"

- P 12, para 3: The sentence beginning with "These results support the possibility ….." should be included in discussion, not in results section. 

- Even a non-native speaker recognizes linguistic deficiencies in the present manuscript. Please revise the manuscript with the help of a native speaker. 

- Page 13: Table1: Were the samples taken from the periphery or the center of the cornea?

Reviewer #3: The work under review attempts to elucidate the risks of corneal transplantation as a vehicle for SARS-CoV-2 transmission. The work is generally well written and very timely, as fear of COVID-19 due to corneal transplantation has slowed use of an urgently needed restorative eye surgery across the globe. Although given the realities of the current pandemic, it is difficult to imagine that eye banks would not test for the presence of virus, whether it be in the donor's nasopharynx, the donor cornea, or both, the current study seeks to clarify the issues with experimental data so that a rational strategy for corneal procurement can be developed.

Some of the work is not novel. Others have assessed the presence of ACE2 and its co-receptors on the ocular surface and found similar results. The novelty of the paper is that the authors attempted to infect corneas ex vivo, which to this reviewer's knowledge, has not been done before. An additional strength of the study is the use of quantitative PCR

Comments: 

- Page 3: regarding the macaque study, neither monkey developed conjunctival findings or showed any evidence for replication in the conjunctiva after topical ocular replication. Most likely the mild COVID-19-like respiratory disease seen in the two macaques was due to nasolacrimal drainage of virus.

- Page 3: "whereas 12% of COVID-19 patients presented ocular symptoms" (ref 9). It is a shame to quote this reference in which 12% of critically ill COVID-19 cases had some eye finding, in most "congestion". There was no control group in this study. It is expected that a fair percentage of critically ill patients with other disorders would also have eye findings, most not directly related to their underlying disease.

- Page 6: the authors might better explain the method of ex vivo infection, which seems nonstandard.

- Page 7 " positive strand represented end products of viral." Please finish the sentence

- Page 12: it is unclear to this reviewer what the relevance is for totaling the positive and negative strand PCR products, to show significance, when productive replication of viral RNA would be indicated by the former and not the latter.

- Page 14: I do not see the data supportive of the first sentence in the Discussion, that "SARS-CoV-2 replication was detected in fresh human cornea and conjunctiva" (or on page 14: "we documented for the first time ...". That would require evidence of production of infectious viral particles in these tissues, whether by increasing culture titers, immunohistochemistry for intact viral particles, or electron microscopic evidence of intact viral particles.

- Page 15: again regarding the macaque study, the authors of that study found viral RNA at 1 day post conjunctival inoculation, not virus, and it was gone thereafter. This would not be consistent with viral replication in the conjunctiva.

Page 15: regarding "SARS-CoV-2 replication after 24h was moderate, with only total viral RNA showing a significant increase", an increase in total viral RNA without increased positive strand RNA is not equivalent to replication of the genome, and may reflect only an abortive process. 

Page 16: " were associated with effective and moderate replication of the virus after experimental infection of freshly excised human corneas". The data supplied does not support this statement. 

This reviewer agrees however with the final sentence and conclusion of the paper.

[LINK]

---

## [Decision Letter · Decision Letter 2]

12 Jan 2022

Dear Dr. Thuret,

Thank you very much for re-submitting your manuscript "Exploration of the ocular surface infection by SARS-CoV-2 and implications for corneal donation: an ex vivo study" (PMEDICINE-D-21-02421R2) for review by PLOS Medicine.

I have discussed the paper with my colleagues and the academic editor and it was also seen again by two reviewers. I am pleased to say that provided the remaining editorial and production issues are dealt with we are planning to accept the paper for publication in the journal.

[LINK]

We look forward to receiving the revised manuscript by Jan 19 2022 11:59PM.   

Sincerely,

Caitlin Moyer, Ph.D.

Associate Editor 

PLOS Medicine

plosmedicine.org

Requests from Editors:

1. Response to reviewers: As suggested by Reviewer 2, please do include the additional experiment to assess whether infected tissue (Study 2) contained replicable SARS-CoV-2 virus.

2. Title: Please capitalize the first word of the subtitle, and please update the title in the manuscript submission system accordingly: “Exploration of ocular surface infection by SARS-CoV-2 and implications for corneal donation: An ex vivo study.”

3. Page 1-2: Please remove the sections titled “Grant” and “Author contributions” from the main text. Please ensure this information is completely and accurately entered into the manuscript submission system.

4. Abstract: Lines 55-58: We suggest removing the numbering from the objectives: “The aims of

this study are to specifically assess the capacity of human cornea to be infected by SARS-CoV-2 and promote its replication ex vivo, and to evaluate the real-life risk of corneal contamination by detecting SARS-CoV-2 RNA in corneas retrieved in donors diagnosed with COVID-19 and non-affected donors.”

5. Abstract: Methods and Findings: Please remove the numbering from the description of the methods. At line 60 we suggest: “To assess the capacity of human cornea to be infected by SARS-CoV-2, the expression pattern of SARS-CoV-2 receptor ACE-2 and activators TMPRSS2 and Cathepsins B and L in ocular surface tissues from non-affected donors was explored…” Please revise similarly for parts 2 and 3.

6. Abstract: Line 64: Please define “TCID” in the text.

7. Abstract: Line 71-72: Please note that the spatial patterns of expression were the derived from the immunohistochemistry experiments.

8. Abstract: Line 73: Please clarify “IP4 strands” for non-specialists.

9. Abstract: Line 73-77: Please clarify if the comparison is between 24 hr and 30 minute time points.

10. Abstract: Line 84: Please remove “a significant” from the sentence.

11. Author summary: Please format the author summary into 3 sections each with 3-4 bulleted points. For example, we suggest:

Why Was This Study Done?

-Corneal transplantation is by far the most common transplantation procedure in the world and there is a severe lack of corneal donation to respond to the increasing patient need.

-The persistent COVID-19 pandemic has dramatically worsened this situation by limiting the access to corneas during a long period of the pandemic and finally increasing the long list of contraindications to corneal donation based solely on an exacerbated precautionary principle.

-Exploration of the SARS-CoV-2 risk of transmission through corneal transplantation is mandatory to adapt the donor selection strategy and improve corneal tissue procurement in the persisting COVID-19 crisis.

What Did the Researchers Do and Find?

-We explored and observed the expression of 4 proteins related to molecular pathways allowing SARS-CoV-2 cellular entry at the ocular surface (cornea, bulbar conjunctiva, 10 samples). These proteins were differentially located in the studied tissues and their expression was variable among donors.

-The experimental infection of five freshly excised corneas with highly concentrated SARS-CoV-2 viral solution promoted only moderate and variable viral RNA multiplication in corneal samples. Twenty-four hours after inoculation, a significant increase in total and positive strand viral RNA was observed in the epithelium of central cornea (p-value: 0.0039 for and 0.0117 respectively) and in corneoscleral rim (p-value: 0.0039 and 0.0391 respectively).

-We explored the presence of SARS-CoV-2 viral genetic material in corneas and respective storage media from donors tested positive or negative to COVID-19 PCR on nasopharyngeal swab. All samples from the 159 donors not contaminated by COVID-19 were negative 111 for SARS-CoV-2 RNA. SARS-CoV-2 RNA was detected in only 3 corneas out of 28 corneas (11%) from donors diagnosed with COVID-19.

What Do These Findings Mean?

-Our results support that the ocular surface expresses in a variable way the proteins involved in SARS-CoV-2 cellular entry and infection. We observed a moderate and variable increase in total and positive viral RNA after experimental infection of human freshly excised corneas. A very low rate of positivity in donors diagnosed with COVID-19 was observed, supporting a low risk of SARS-CoV-2 presence in donor corneas.

-This work suggests that it would be beneficial to procure corneas from donors tested positive to COVID-19 (nasopharyngeal sawbs), test their corneal storage media and transplant the negative corneas.

12. Introduction: Line 135-136: We suggest revising to: “Finally, it deprives some patients and/or families of access to tissue donation, which can be a beneficial aspect of the mourning process…” or similar, depending on your meaning.

13. Methods: Line 249: Please clarify (H0) in the text.

14. Methods: Line 284: Please define the abbreviation “TCID” in the text.

15. Methods: Line 323: We suggest including a reference for the online PCR protocol, rather than inserting the link into the text.

16. Methods: Line 388-389: PLOS does not permit "data not shown.” Please remove this claim, or do one of the following: If you are the owner of the data relevant to this claim, please provide the data. If the data not shown refer to a study from another group that has not been published, please cite personal communication in your manuscript text (it should not be included in the reference section). Please provide the name of the individual, the affiliation, and date of communication. The individual must provide PLOS Medicine written permission to be named for this purpose.

17. Methods: Line 410-412: For each result presented (e.g. in the respective table legend) please note the test that was used.

18. Discussion: Line 533: Please revise to “may not be a good viral vector” or similar.

19. Discussion: Line 636: We suggest tempering this slightly “Our results suggest that only corneas with positive PCR…”

20. References: For reference 4 and 5, if these are online materials, please provide the weblink, and please check that formatting is correct: Please use the "Vancouver" style for reference formatting, and see our website for other reference guidelines https://journals.plos.org/plosmedicine/s/submission-guidelines#loc-references

Please check formatting of reference 11.

21. Figure 1: Please define “OC” in the legend.

22. STROBE Checklist: Thank you for including the checklist. Please include the checklist as a separate file (e.g. S1 Checklist). Please revise the checklist to remove any references to page numbers. Please refer only to sections and paragraph numbers.

Comments from Reviewers:

Reviewer #2: The authors addressed all major points and performed additional experiments as requested. The manuscript improved substantially. 

Notably, clear evidence of replicable virus in corneas in study 2 could not be shown by the authors despite the well performed additional experiments. However, the authors improved the discussion accordingly and pointed out the possibility of an abortive process. Taken together, the data supports the conclusions and the demand to question current donor selection algorithms in view of the corneal donor shortage. I would suggest including the additional experiment in the paper, in my opinion it gives relevant additional information and can contribute to reduce "positive publication bias". 

Taken together, the paper represents an important contribution to the present debate on donor selection and eye bank procedures in the COVID 19 pandemic, I recommend to publish now without further changes.

[LINK]

---

## [Editor Report · Decision Letter 3]

19 Jan 2022

Dear Dr Thuret, 

On behalf of my colleagues and the Academic Editor, James Chodosh, I am pleased to inform you that we have agreed to publish your manuscript "Exploration of the ocular surface infection by SARS-CoV-2 and implications for corneal donation: An ex vivo study" (PMEDICINE-D-21-02421R3) in PLOS Medicine.

Please also address the following editorial requests:

-Introduction: Line 118: Please specify the year in this sentence: “Nevertheless, these recommendations became obsolete at least in France in November…”

-Methods: Line 215: Please reference the corresponding supporting information file here.

-Methods: Line 352: Please remove the trademark symbol from: “...nuclei were stained with TO-PRO™-3 Iodide…”

-Methods: Line 390: Please reference the corresponding supporting information file here: “Consistency between the two extraction kits was experimentally validated prior to sample processing (data provided in supporting information).”

PRESS

Sincerely, 

Caitlin Moyer, Ph.D. 

Associate Editor 

PLOS Medicine